# Application Research of Biochar for the Remediation of Soil Heavy Metals Contamination: A Review

**DOI:** 10.3390/molecules25143167

**Published:** 2020-07-10

**Authors:** Sheng Cheng, Tao Chen, Wenbin Xu, Jian Huang, Shaojun Jiang, Bo Yan

**Affiliations:** 1SCNU Environmental Research Institute, Guangdong Provincial Key Laboratory of Chemical Pollution and Environmental Safety & MOE Key Laboratory of Theoretical Chemistry of Environment, South China Normal University, Guangzhou 510006, China; ShengC@m.scnu.edu.cn (S.C.); 18371807641@163.com (J.H.); shaojunj93@163.com (S.J.); bo.yan@m.scnu.edu.cn (B.Y.); 2School of Environment, South China Normal University, University Town, Guangzhou 510006, China; 3Dongjiang Environmental Company Limited, Nanshan District, Shenzhen 518057, China; xuwb@dongjiang.com.cn

**Keywords:** biochar, pyrolysis, heavy metals, soil remediation, bioavailability

## Abstract

Soil contamination by heavy metals threatens the quality of agricultural products and human health, so it is necessary to choose certain economic and effective remediation techniques to control the continuous deterioration of land quality. This paper is intended to present an overview on the application of biochar as an addition to the remediation of heavy-metal-contaminated soil, in terms of its preparation technologies and performance characteristics, remediation mechanisms and effects, and impacts on heavy metal bioavailability. Biochar is a carbon-neutral or carbon-negative product produced by the thermochemical transformation of plant- and animal-based biomass. Biochar shows numerous advantages in increasing soil pH value and organic carbon content, improving soil water-holding capacity, reducing the available fraction of heavy metals, increasing agricultural crop yield and inhibiting the uptake and accumulation of heavy metals. Different conditions, such as biomass type, pyrolysis temperature, heating rate and residence time are the pivotal factors governing the performance characteristics of biochar. Affected by the pH value and dissolved organic carbon and ash content of biochar, the interaction mechanisms between biochar and heavy metals mainly includes complexation, reduction, cation exchange, electrostatic attraction and precipitation. Finally, the potential risks of in-situ remediation strategy of biochar are expounded upon, which provides the directions for future research to ensure the safe production and sustainable utilization of biochar.

## 1. Introduction

Soil is the final destination of heavy metals (HMs) whether they are from natural or anthropogenic sources. Mineral resource exploiting and smelting [1], metal electroplating [2], paint and coating processing [3], electronic equipment manufacturing [4], farmland sewage irrigating [5] and pesticide and chemical fertilizer nonstandard applying [6] are the primary anthropogenic activities that aggravate HMs contamination in soil. For instance, more than 30,000 tons of chromium and 800,000 tons of lead have been released into the environment globally in the past half century, most of which eventually accumulates in soils [7]. It is reported that approximately one-sixth of the total agricultural land area in China and about 600,000 hectares of brown field sites in America have been contaminated by HMs [8]. Hitherto, cadmium, lead and arsenic pollution, and associated ecological health risks in southeast China, are more severe than those in northwest China; similarly, those in industrial regions are worse than those in agricultural regions [7]. According to the results of a national soil survey [9] in 2014, the over-standard rates of cadmium, mercury, arsenic, copper, lead, chromium, zinc and nickel in soils of China are 7.0%, 1.6%, 2.7%, 2.1%, 1.5%, 1.1%, 0.9% and 4.8%, respectively. Soil HMs enter the food chain mainly through agricultural crops [10] and ultimately accumulate in organisms through diet, respiratory inhalation, skin contact and other exposure pathways, which directly or indirectly cause serious negative effects on human health [11,12]. The occurrence of cancer villages caused by HMs are the most immediate warning. Soil degradation and reduction of agricultural production land caused by HMs pollution brings an urgent need for the application of various efficient in-situ and ex-situ remediation techniques, to lessen the ecological risk of HMs and maximize the quality and security of agricultural land.

Over recent years, physical remediation (washing, thermal desorption, solidification and guest land methods), chemical remediation (vitrification, leaching, immobilization and electrokinetic methods), and biological remediation (microorganisms and plants) approaches have been applied to achieve this objective [13]. Nevertheless, these methods more or less exist with respective limitations i.e., complicated technique, inefficiency, poor feasibility, short duration, high economic cost, high secondary risk and so on [14]. At present, applying amendments to HM-contaminated soil is considered to be one of the most promising in-situ remediation techniques [15]. The frequently used soil additions include phosphate compounds, liming materials, clay minerals, coal fly ash, organic composts, metal oxides, and biochar [16,17]. In brief, the immobilization process of HMs can be achieved mainly through adsorption, complexation, reduction, and precipitation reactions, which cause the redistribution of HMs from soil liquid phases to solid phases, so as to reduce their mobility and bioavailability [16].

Biochar is a kind of carbon-rich, porous substance with abundant active organic functional groups and carbon aromatic structures with a neutral to alkaline pH value, relatively high cation exchange capacity, large specific surface area, and negative surface charge [18,19]. Numerous researchers report that the seed germination [20], plant growth [21,22,23], crop yields [23,24], and microbial activity and population [22,25,26] have been significantly increased in the HM-contaminated soil amended with biochars. Meanwhile, the effects of biochar on the immobilization/mobilization for different kinds of HMs have been confirmed in many pot experiments and field trials [21,27,28,29]. Furthermore, the production process of biochar is regarded as an efficient management method to dispose of a large number of organic wastes, which shows certain advantages in economic benefits and feasibility aspects. Remarkably, it cannot be completely ignored that there are still some potential risks in the field application of biochar, and these potential threats may hamper its further application. On this basis, the following aspects of biochar are reviewed: 1) preparation technologies, performance characteristics and influencing factors; 2) interaction mechanisms with different kinds of HMs; 3) effects on plant growth and HM bioavailability; and 4) potential risks in field engineering applications. In addition, the promotion and application of biochar in the future are also discussed.

## 2. Preparation of Biochar

The feedstock for biochar preparation mainly includes wood chips/branches [30,31,32], agricultural residues [21,33,34] and other woody biomass, as well as animal manure [35,36,37,38], sewage sludge [39,40,41] and other organic wastes. As shown in Table 1, thermochemical conversion technologies involved in the preparation process usually include fast pyrolysis, intermediate pyrolysis, slow pyrolysis, gasification, hydrothermal carbonization, torrefaction, etc. [42,43]. These technologies are mainly classified based on different heating rates, peak temperatures, residence times, reaction atmospheres and other parameters, where biochar, bio-oil and syngas are the main products. The yield of biochar is seriously affected by different operating conditions; several common thermochemical conversion technologies and their approximate product yields are discussed in Table 1.

Fast pyrolysis can be defined as the thermochemical decomposition process of biomass with low energy density at a moderate pyrolysis temperature in the presence of little or no oxygen. Due to the characteristics of higher heating rate (> 200 K min^–1^) and shorter residence time (< 2 s) in this process, bio-oil with high energy density, syngas with relatively low energy density and a small amount of biochar can be obtained (Table 1) [52]. On the contrary, slow pyrolysis is the common type of pyrolysis which is conducive to the formation of biochar rather than the generation of liquid and gaseous products. In this process, the biomass is pyrolyzed in a wide range of carbonization temperature with a heating rate of about 0.1~1 ℃ s^–1^ for a residence time between few hours and even days [52]. During slow pyrolysis, the fixed carbon content of biochar may increase with the rise of peak temperature, which is particularly prominent in the range of 400~500 °C [44]. The operating conditions of intermediate pyrolysis are between fast pyrolysis and slow pyrolysis, which better balances the distribution of solid–liquid product yield. The biochar produced by this pyrolysis mode has a brittle structure and does not contain a high quantity of reactive tar, which is suitable for the application of solid fuel, soil amendments and fertilizer [46].

Gasification is a process of direct contact oxidation of dry biomass with air, steam, oxygen, nitrogen, carbon dioxide or a mixture of these gases [43]. The primary product of gasification is a combustible gas which is packed with H_2_, CO and CH_4_, while the biochar with low yield contains a high amount of toxic substances such as polyaromatic hydrocarbons, and alkali and alkaline earth metals, that are attributed to the result of high-temperature reactions [51]. Hydrothermal carbonization is performed under a given pressure (2~6 MPa) and temperature (180~300 °C) that the feedstocks do not require for drying pretreatment, which is usually some wet biomass or dry biomass mixed with water. Compared to the pyrolysis and gasification processes, hydrothermal carbonization biochar (hydrochar) shows several advantages [51,53]. For instance, the hydrochar is characterized by high yield and high purity, and possesses a higher degree of aromatization and more surface functional groups. In addition, hydrochar contains a lower alkali and alkaline earth and heavy metal content, and a higher carbon content and heating value. Torrefaction, also referred to as mild pyrolysis, results in approximately 30% mass loss of biomass. The main products of torrefaction are organic carbon compounds with high specific energy density, but these cannot be referred to as a “biochar”, because torrefaction is just the previous section of the pyrolysis process [43,51]. Consequently, the physicochemical properties of the torrefaction product is between that of biochar and biomass, and it also remains some volatile organic compounds.

Pyrolysis is a conventional process for the preparation of biochar. In this review paper, the properties of pyrolyzed biochar are preliminarily discussed, and the close connection between biochar performance characteristics and biomass feedstock species, pyrolysis temperature and residence time are clarified. According to the above factors (different preparation conditions), the performance characteristics of biochar are reviewed in detail in the next chapter, which provides a theoretical foundation for explaining the interaction mechanisms between biochar and soil HMs.

## 3. Performance Characteristics of Biochar

### 3.1. Elemental Composition

Element composition and content of biochar are a function of biomass species and pyrolysis temperature [54]. Generally, the content of total N, P, K, Ca, Mg and other nutrient elements in biochar prepared from poultry manure is higher than that of woody biomass, while the content of total C is the opposite [55]. Meanwhile, poultry manure is rich in mineral elements like K and P, which are important for plant growth, and thus the poultry-manure-derived biochar may be suitable as an ideal soil amendment instead of fertilizer [56].

The rise in pyrolysis temperature of biomass commonly results in the increase of ash and C content of biochar [50,57]. The N content of lignocellulosic type biochar increased slightly with the rise of pyrolysis temperature [58,59], while that of animal-manure- and sewage-sludge-derived biochar shows a downward trend [35,60,61,62]. Furthermore, the pyrolysis conditions of biomass such as a higher temperature and longer residence time are beneficial to the accumulation of total P and K [50,63], the release of Ca, Mg and Si, and the retention of Fe, Mn and S [47]. Correspondingly, with the increase in reaction time and temperature, other unstable substances (containing H and O elements) of biomass are removed by deoxygenation, dehydration and decarboxylation reactions progressively, which leads to the loss of volatile organic compounds and the reduction of H/C–O/C ratio [51,57,64,65]. These results indicate that the high-temperature biochar more easily forms a very stable crystal graphite-like structure and possesses a higher carbonization degree and more aromatic structures [66,67,68]. For example, Jindo et al. [66] reported that the O/C ratio of biochar pyrolyzed in the temperature range of 400~500 °C changed according to the following order: rice straw > rice husk > wood chips of apple tree > wood chips of oak tree. Therefore, these results indicate that there is a higher content of lignin and a slower mineralization rate in woody biomass (i.e., apple tree, oak tree), compared with herbaceous biochar (i.e., rice straw, rice husk) and sewage sludge biochar; furthermore, it has a lower O/C ratio, making woody biochar’s structure more stable [67].

### 3.2. Functional Groups Abundance

As mentioned in Section 3.1, a high pyrolysis temperature commonly results in the decrease of the H/C, O/C and N/C ratios in biochar, which immediately indicates the decrease of its abundance of hydroxyl, carboxylic and amino functional groups [18]. Chen et al. [57] summarized the variation of FTIR characterization of pine wood shaving derived biochar with pyrolysis temperature: 1) for 150 °C, biochar is rich in -OH groups, CH_2_ units, C=O, C=C, aromatic CO-, and phenolic-OH; 2) for 250 °C, C=O and C=C stretching vibrations were enhanced; 3) for 350 °C, the band CH_2_ units disappeared completely, and aromatic ring and C=C stretching vibration of lignin strengthened; 4) for 500 °C, C=O and C=C stretching vibrations were significantly weakened; 5) for 700 °C, merely C=C of lignin and aromatic C-H vibrations were spotted. During the whole heating process, the band C-O-C of cellulose and hemicellulose reduced with the rise of pyrolysis temperature, until it disappeared. Hence, these results show that hemicellulose with short side chains, thermally stable cellulose, and lignin with phenolic structures contained in lignocellulosic are generally decomposed at 200~350 °C, 305~375 °C and 250~500 °C, respectively [69]. In addition, Ding et al. [70] indicated that compared with 250 °C pyrolyzed sugarcane-derived biochar, C≡C and C=O in 500 °C pyrolyzed biochar were relatively reduced, while C-O completely disappeared.

Moreover, the abundant surface functional groups such as C-O, C=O, -COOH and -OH in biochar possess high modifiability, which is the foundation for the preparation of various functionalized carbon materials [52]. For instance, Yang et al. [71] incubated walnut-shell-derived biochar with Al, Ca, Fe minerals or kaolinite, and the modified biochar’s relative content of C-C, C=C and C-H increased from 63.8% to 72.5~81.8%, while C-O, C=O, and -COOH decreased from 36.3% to 16.6~26.5%. This result means that the interaction between biochar and minerals (Al, Ca, Fe, or kaolinite) has prevented the oxidation of C-C, C=C, C-H into C-O, C=O or -COOH, which enhanced the oxidation resistance of biochar surface. This is related to the modification process of biochar, which has been reviewed by previous scholars in the following papers [18,31,72], and will not be further analyzed in this paper.

### 3.3. Cation Exchange Capacity (CEC) and Specific Surface Area (SSA)

With the pyrolysis processes conducting, the SSA (specific surface area) value of biochar has significantly increased compared to the feedstock, while the relatively low temperature pyrolyzed biochar has the highest CEC (cation exchange capacity) value [73]. Although high-temperature biochar cannot possess the highest CEC and SSA values simultaneously, there are adequate functional groups remaining in the biochar structures to provide negative charges. The low O/C atomic ratio of high-temperature biochar results in a decrease in the CEC value, which is mainly manifested by the reduction of volatile organic compounds and acidic functional groups [74]. In other words, the high SSA and pH value of biochar pyrolyzed at higher temperatures (> 600°C) may compensate for the low CEC value to supply greater CEC provision to soil [54]. Thereby, biochar is a combination of charged surface functional groups and specific surface area, which can combine with HMs by adsorption and complexation reactions. For example, Yuan et al. [75] raised the pyrolysis temperature of sewage sludge from 300 °C to 700 °C, and found that the O/C ratio of biochar decreased from 0.33 to 0.05 and the volatile matter content reduced from 27.4% to 5.5%, but the SSA value increased from 14.37 m^2^ g^−1^ to 26.70 m^2^ g^−1^. Similarly, Heitkötter et al. [76] used corn digestate (derived from maize silage) as feedstock, as the temperature increased from 400 °C to 600 °C, the CEC value decreased by 29.9%, but the SSA value increased by 50.7% which exactly made up for the deficiencies in the CEC value reduction. The cation exchange capacity of biochar was enhanced by the tendency of attracting positive charges through its surface functional groups, which is an important feature for the remediation of HM-contaminated soil [77]. Furthermore, the porosity and pore size of biochar still depend on pyrolysis temperature, because the release of volatile organic compounds at higher pyrolysis temperatures may promote the formation of micropores.

### 3.4. pH Value

The pH value of biochar is mostly alkaline, and normally increases with the rise of pyrolysis temperature, which means biochar possesses the abilities to improve soil pH and CEC value, and to reduce soil acidity and bioavailability of certain HMs [78,79,80]. Fidel et al. [81] summarized four broad categories of biochar alkalinity, including: (1) surface organic functional groups; (2) soluble organic compounds; (3) carbonates/bicarbonates; and (4) other inorganic alkalis such as oxides, hydroxides, sulfates, sulfides, and orthophosphates. Surface organic functional groups have a long-term effect on the amelioration of soil properties (e.g., pH and CEC value), while soluble organic and inorganic alkalis contribute to the short-term improvement of soil acidification [54,81]. The functional groups separated from the pyrolysis of biomass are predominantly acidic in essence, such as the carboxyl group, hydroxyl group, or formyl group [73]. For others, the alkalinity of remaining solid (includes ash) raised with the increase number of functional groups released by biomass, therefore, the increase of pH value of biochar is the direct result of the increasing degree of carbonization [73]. The carbonates formed by mineral elements were considered to be the primary alkaline substances in biochar, and especially the biochar pyrolyzed at high temperatures possesses a higher content of carbonates a stronger buffer capacity [80]. Shen et al. [34] reported that the pH value (7.94) of rice-straw-derived biochar pyrolyzed at 300 °C is alkalescence, while the biochars with higher pH values (10.40 and 10.68) can be obtained at higher pyrolysis temperatures (500 °C and 700 °C). The results are attributed to the decomposition of acidic functional groups such as the carboxyl group and phenolic hydroxyl group, and the formation of alkaline minerals like K_2_O in high-temperature pyrolyzed biochar. In addition, there is a high ash content in poultry manure and algae biomass, so the pH value of biochar pyrolyzed at the same temperature is higher than that of other woody biochar [82]. However, in some studies, the biochar produced by hydrothermal carbonization is typically acidic [83]. For example, the pH value of *Miscanthus*-derived hydrochar prepared by Gronwald et al. [84] at 200 °C is 3.8, and Cui et al. [85] found that the pH values of *Hydrocotyle verticillata*-, *Myriophyllum spicatum*- and *Canna indica*-derived hydrochar (200 °C) are 5.07, 4.97 and 6.48. Liu et al. [86] adjusted the pH value of the initial solution (pH = 2, 3, 5, 7, 9, 11, 12) of the hydrothermal carbonization process, so that the pH value of the prepared sewage-sludge-derived hydrochar was still weakly acidic or neutral (corresponding pH = 5.05, 6.11, 7.24, 6.60, 6.62, 6.74, 6.94). The presence of carboxyl functional groups on the surface of hydrochar as a result of formation of acetic and formic acids during the hydrothermal carbonization process could be the reason for the low pH value [87].

## 4. Remediation of Soil HMs Contamination by Biochar

### 4.1. Interaction Mechanisms of Biochar and HMs in Soil

The different sources of biomass feedstocks, and the diverse pyrolysis conditions applied in the preparation processes, lead to various biochar performance characteristics, which may in turn affect the interaction mechanisms between biochar and HMs. On the other hand, the greatest concerns of HMs have been focused on copper, arsenic, cadmium, lead, mercury, and chromium. Table 2 summarizes the research progress on biochar applications for the remediation of HM-contaminated soil, which included different types of biochar, different application conditions, and different HM treatment efficiencies. The various mechanisms proposed for the interaction of biochar with HMs are summarized in Figure 1. It shows that the abundant surface functional groups, mineral substances, alkaline metal ions, π-electrons, organic matters, and pore structures of micropores provided by biochar are the effective binding sites of HMs. Biochar is able to absorb or combine soil HMs through complexation, reduction, cation exchange, electrostatic attraction, and precipitation functions, or convert HMs from inorganic states into organic states, which changes HM mobility and bioavailability [14,88,89], and then improves soil agronomic benefits. Therefore, the interaction mechanisms between biochar and HMs are critical for the soil remediation and are discussed in detail in the following sections.

#### 4.1.1. Copper (Cu)

As described by Meier et al. [36], the functional groups (especially for -OH) and negative ζ-potential existing in chicken-manure-derived biochar has been proved with high affinity for Cu. The immobilization process of Cu is achieved by increasing soil pH value and inducing the liming effect, to stimulate the complexation of Cu(Ⅱ) with biochar surface functional groups (e.g., C=O and phenolic-OH). Additionally, Rechberger et al. [30] found that carbonates and hydroxides in the ash of woodchip-derived biochar are the important adsorbents for Cu(Ⅱ), which are able to promote the formation of CuCO_3_ and Cu(OH)_2_, and this is also illustrated by the bamboo-derived biochar prepared by Zhang et al. [93]. Therefore, the essential point of Cu immobilization is to use the alkalinity of biochar to improve soil pH value [92]. On the other side of the shield, the mobility of Cu is highly affected by the content of soil-dissolved organic carbon (DOC). For instance, Park et al. [91] introduced chicken-manure-derived biochar to Cu-spiked soil, which led to the increase of soil DOC content and provoked the conversion of Cu(Ⅱ) into Cu complexes with higher solubility. Wagner et al. [106] also reported that the *Miscanthus*-derived biochar can increase the Cu concentration in soil solution. In other words, the increase of Cu concentration in soil pore water is the proximate consequence of the Cu(Ⅱ) desorption from soil with the form of organic complex. Hence, the immobilization/mobilization of Cu by biochar should be further studied according to the actual types of biochar.

#### 4.1.2. Arsenic (As)

Phosphorus (P) and As have similar chemical properties and the soil P content is a critical factor in controlling the mobility of As [25]. There is a significant positive correlation between phosphate content and arsenate content in As-contaminated soils. After the soils are treated with P-containing biochar, the competition between soluble phosphate and arsenate that occurred on the adsorption sites of soil particles promoted the desorption of As from soil solid phase, and increased the As concentration in pore water [45], such as soybean-stover-, pine-needle- [25], and rice-straw-derived biochar [94]. The role of soil DOC is similar to P [95], but DOC has another effect. For example, Wang et al. [26] indicated that the application of rice-straw-derived biochar under the anaerobic conditions increased the abundance of Fe-reducing bacteria (e.g., *Clostridum*, *Bacillus* and *Caloramator*) in paddy soil, promoted the reduction of As(Ⅴ) adsorbed on the amorphous Fe/Al oxides. To be brief, the increase of DOC content in soils enhanced the microbial reduction effect of As(Ⅴ), and finally stimulated the release of As (Ⅲ) from paddy soil. Under the anoxic conditions, similar results were obtained in the application of oil palm fiber derived biochar prepared by Qiao et al. [96]. Biochar, with high aromaticity and alkalinity, and which can act as an electron shuttle, likes humus to promote the microbial reduction of Fe(Ⅲ) and As(Ⅴ) simultaneously [26,96]. In addition, Choppala et al. [27] reported that the π-electrons provided by the functional groups on the surface of chicken-manure-derived biochar could promote the reduction of As(V), which is another important factor to enhance the mobility of As. However, the Fe-biochar prepared by Yin et al. [94] and Mn-biochar prepared by Yu et al. [23] are able to adsorb As to the Fe/Mn oxides of biochar surface, thus commendably limiting the migration of As into soil solution, which proved the effectiveness of As immobilization by the modified biochar (not listed in Table 2).

#### 4.1.3. Cadmium (Cd)

The activity of Cd in soils strongly depends on soil pH value [99]. The alkaline substances such as CO_3_^2−^, PO_4_^3−^ and OH^−^ contained in biochar commonly have strong adsorption and binding capacity to Cd in soils, which makes the free Cd(Ⅱ) transform into Cd(OH)_2_, Cd_3_(PO_4_)_2_ and CdCO_3_ precipitates [30]. The high adsorption affinity produced by the cation exchange effects of soil calcite (CaCO_3_) with Cd(Ⅱ) is the main factor to reduce the bioavailability of Cd; of course, the abundant functional groups and large specific surface area in biochar are also critical for Cd immobilization [29]. For instance, Yin et al. [94] used 1~2% rice-straw-derived biochar to treat farmland soil in a mining area, and found that the Cd concentration in pore water of soil rhizosphere was significantly reduced, and the corn-straw-derived biochar prepared by Gao et al. [99] decreased 91% of the CaCl_2_-extractable Cd content in the farmland soil. Besides, Bian et al. [98] applied 40 t ha^−1^ wheat-straw-derived biochar to paddy soil, and the CaCl_2_-extractable Cd and DTPA-extractable Cd was reduced by 59% and 24%, respectively, over the past three years. Thereby, the result of biochar in increasing soil pH value is extremely effective for the immobilization of Cd. It is worth mentioning that the basic substances in biochar can also stimulate the deprotonation of acid functional groups of biochar, and further enhance the adsorption capacity of Cd [107].

#### 4.1.4. Lead (Pb)

The immobilization process of Pb in soils by biochar is relatively simple. For example, Ahmad et al. [25] considered that the immobilization process of Pb stimulated by soybean-stover-derived biochar is attributed to the π-cation electron donor–acceptor interaction, which occurs by the π-electron-rich biochar graphene surface and π-electron-deficient positively charged Pb(Ⅱ) ion. Furthermore, the cation exchange and precipitation reactions between basic substances (CO_3_^2−^, OH^−^ and other alkaline earth Ca^2+^, Mg^2+^) in biochar and Pb(Ⅱ) are able to achieve a significant immobilization effect of Pb, such as the formation of Pb_3_(CO_3_)_2_(OH)_2_ precipitation. In particular, the poultry-manure-derived biochar [38] and the sewage-sludge-derived biochar [39] contain abundant phosphates, which are able to form insoluble compounds, such as Pb_5_(PO_4_)_3_Cl, Pb_5_(PO_4_)_3_OH and β-Pb_9_(PO_4_)_6_, with Pb(Ⅱ) to reduce the mobility of Pb [25,101]. The functional groups also have certain effects on the immobilization of Pb. Igalavithana et al. [100] reported that the vegetable-waste-derived biochar not only improved soil pH value, but also promoted the immobilization of Pb by the strong covalent bonding action of N-containing functional groups (especially for -NH_2_) on the biochar surface, thereby effectively decreasing the concentration of NH_4_OAc-extractable Pb.

#### 4.1.5. Mercury (Hg)

Hg, as a special metal, has great toxicity in soil, but biochar is an effective tool for soil remediation. As Wang et al. [31] remarked, the carboxyl group in a hardwood-derived biochar surface and soil Hg(Ⅱ) ion develop a coordination reaction generating a complex of -COOHg^+^ precipitate, thereby reducing the mobility of Hg and the thiol functionalities and sulfoxide groups are also able to react with Hg(Ⅱ) ion to form -S(Hg)- and [Hg(OSR_2_)_6_^2+^] precipitates. Additionally, Xing et al. [28] found that rice-husk-derived biochar possesses a higher sulfate concentration compared with wheat-straw-derived biochar, which is more effective to promote the mercury–sulfur coordination reaction and to produce sulfides precipitation. It is extremely important to know that methylation of Hg is a special environmental biogeochemical behavior, and once the inorganic-Hg in soils is converted into methylmercury (MeHg), its toxicity and bioaccumulation will be enormously enhanced [108]. The release of rice root exudates reduced soil pH value, thereby enhancing the methylation of Hg, while the application of alkaline biochar increases soil pH value and effectively inhibits methylation. Zhang et al. [41] reported that sewage-sludge-derived biochar with high organic matter content can stimulate the growth and activity of heterotrophic microorganisms in Hg-contaminated acidic farmland soil, thus promoting the formation of MeHg. However, the utilization rate of MeHg in rice has been significantly decreased, which led to the inhibition of MeHg accumulation in rice and effectively reduced the bioavailability of organic-Hg [41].

#### 4.1.6. Chromium (Cr)

Firstly, it should be pointed out that Cr exists in soils in the form of two valence states, namely Cr(Ⅵ) and Cr(Ⅲ). Therefore, the immobilization of Cr in soils is a complicated combination process of adsorption–reduction–precipitation. The interaction mechanism between biochar and Cr is mainly manifested in the surface adsorption effect of Cr(Ⅵ) by the oxygen-containing functional groups such as C-O, C=O, -COOH and -OH in biochar, as well as the reduction reaction of electrons provided by biochar [105]. Specifically, biochar transforms Cr(Ⅵ) into Cr(Ⅲ) with lower toxicity and solubility by adsorption–reduction effects, and participates in the formation of Cr_2_O_3_ and/or Cr(OH)_3_ precipitation, so as to achieve the purpose of Cr immobilization [109,110]. On the other hand, Choppala et al. [27] applied chicken-manure-derived biochar to Cr-spiked soil with low organic matter content, which significantly increased the supply of soil organic carbon and nutrients, enhancing the soil respiration and microbial activity effectively and finally showing a superior microbial Cr(Ⅵ) reduction effect and reduced the Cr biotoxicity. Moreover, Mandal et al. [111] found that the Cr(Ⅵ) reduction effect of animal-manure-derived biochar in acidic soil was significantly higher than that in alkaline soil.

In summary, many HMs can be continuously immobilized in soil by biochar through specific or non-specific surface adsorption. It is noteworthy that not only the activity of Cu and As are affected by DOC content in soil, but also Pb and Cd. This phenomenon is attributed to the fact that the high-dose application of biochar in soil results in the sharp increase of DOC content, which leads to the complexation of Pb and Cd with DOC, and finally increases the mobility and bioavailability of Pb and Cd. In addition, the adsorption and cation exchange reactions between HMs and biochar ash (e.g., K^+^, Na^+^, Ca^2+^, Mg^2+^ and other alkali metal ions) are also important factors for the immobilization of HMs.

### 4.2. Effect of Biochar on Bioavailability of HMs in Soil

Plants mainly absorb, transport, and accumulate HMs from contaminated soil by roots. Thus, the primary objectives of biochar soil remediation are limiting the migration and transformation rates of HMs in soil and reducing their bioavailability, so as to prevent HMs from entering organisms through the food chain and to eliminate their toxic effects. The ultimate goal of HM-contaminated soils remediation is to increase crop yields on the premise of ensuring food production safety.

On the one hand, the introduction of biochar provides a source of organic matter, N, P, K, Ca, Mg and other nutrients to the soil, which enhances soil enzyme and microbial activities. On the other hand, the plant root environment, soil water retention and saturated hydraulic conductivity [112] can be improved with the presence of biochar, and plant growth and nutrient absorption can be promoted. Finally, it increases the plants biomass, and dilutes the content of HMs in plant tissues to reduce their phytotoxicity. Meier et al. [36] indicated that 5% chicken-manure-derived biochar can reduce the uptake of Cu from 66.9 mg kg^−1^ to 36.6 mg kg^−1^ in the aboveground part of *Oenothera picensis* plants in copper-mine-polluted soil, and increase the biomass of shoots and roots by 3.5 times and 3.1 times respectively. Xing et al. [28] reported that after applying 24 t ha^−1^ and 72 t ha^−1^ rice-husk-derived biochar to mercury-contaminated farmland soils, the Hg content in rice grains reduced to 10 ng g^−1^ and 7.2 ng g^−1^, which significantly inhibited the transportation of Hg from soil to rice grains, and successfully reached the national standard (below 20 ng g^−1^). Similarly, Li et al. [97] applied 3% of soybean-straw-derived biochar (hydrothermal carbon at 350℃) to the arsenic and cadmium co-contaminated farmland soil, which reduced the bioaccumulation of As in rice plants by 88%. Besides, compared with the control group, As(Ⅲ) content decreased from 3.47 mg kg^−1^ to 0.29 mg kg^−1^, As(Ⅴ) decreased from 715 μg kg^−1^ to 150 μg kg^−1^, and the treatment effects on Cd were similar [97]. As for cadmium, a field trial studied by Zheng et al. [113] showed that, when the application rates of soybean-straw-derived biochar and rice-straw-derived biochar were 20 t ha^−1^, the content of Cd in rice roots, rice shoots, rice husks and rice grains decreased by 25.0~44.1% and 19.9~44.2%, and 46.2%~70.6% and 25.8%~70.9%, respectively. Furthermore, the effectiveness of rice-straw-derived biochar in reducing Cd accumulation in rice grains was also confirmed in the long-term field effects studied by Zhang et al. [29]. Cd contamination of farmland soil is particularly prominent in China, but it happens that China is a large agricultural country, which has a large output of agricultural waste with low cost. Considering the economic value of biochar application, using rice straw, wheat straw and other agricultural wastes to produce biochar and returning it to the field is the best choice.

In a word, the immobilization effects of soil HMs, the limitation of HM uptake and accumulation by plants, the enhancement of plant biomass, and the dilution effect of HMs in plant tissues are the four pivotal performances of biochar to reduce the bioavailability of HMs [114,115]. Although certain HMs can be combined with soil DOC and converted into organic complex forms, which are activated finally, these organic complexes commonly possess stability and are not readily or directly absorbed by plants, which has little impact on HM content in plant tissues.

## 5. Potential Risks

Applying biochar as an addition to soil in-situ remediation should not only consider the HM immobilization/mobilization effects, but also take into account the long-term stability and potential ecological risks of biochar, which similarly depend on the type and performance characteristics of biochar. Figure 2 briefly introduced the advantages and disadvantages of biochar in the remediation of soil HM contamination. Firstly, it is reported that biochar may be the carrier of HMs [116], volatile organic compounds (VOCs) [117], dioxin (PCDD/Fs) and polycyclic aromatic hydrocarbons (PAHs) [118,119] and other toxic substances, and the demands for remediation of HM contamination in soil varies from 1.5 to 72 t ha^−1^ or even higher [24,28,98,113]. Therefore, the toxic substances may be released into the soil/air/water environment with the application of the double-edged biochar, which will pose a secondary pollution and ecological risk. So far, from the perspective of soil, biochar plays an important role in the carbon utilization and non-CO_2_ greenhouse gases emission reduction [104]. However, the research on increasing greenhouse gas emissions by biochar has also been reported, i.e., under specific conditions, the application of biochar can promote the emissions of CO_2_ [120], N_2_O [121], CH_4_ [122] to a certain extent. For example, although biochar can reduce CH_4_ emissions, it may also promote N_2_O emissions, and vice versa, and this depends on the diversity of biochar application conditions. Therefore, the emission reduction effect of biochar cannot be reflected in all types of greenhouse gases, and blind application may cause negative effects, so the biochar–greenhouse gas interaction should be considered in the field application of biochar. Meanwhile, other research has indicated that biochar can inhibit the efficacy of soil pesticides and their biodegradation effects [123], which makes the ability of agricultural weeding and insecticides unable to achieve the expected effects. The residue of pesticides may be related to the strong adsorption and binding capacity of biochar. Furthermore, although biochar can improve the biological activity of bacteria (e.g., *Geobacter*, *Anaeromyxobacter* and *Clostridium*) [26,96], it may bring about a negative impact on the survival, growth and diversity of, for example, acidophilic earthworm and fungi biological communities [124,125].

## 6. Summary and Future Perspectives

It is a feasible strategy of green, economic and environmental protection to apply biochar to remediate the HM-contaminated soil. Several thermochemical conversion technologies have been used to prepare biochar, and the performance characteristics of biochar are highly affected by the type of feedstock materials, pyrolysis temperature, and residence time. The immobilization or mobilization mechanisms of HMs includes complexation, reduction, cation exchange, electrostatic attraction and precipitation reactions. The HM uptake and accumulation of plants can be suppressed by biochar through the variation in soil pH value, DOC content, and other alkaline mineral substances content, and achieve the goal of promoting plant growth, reducing HM bioavailability and improving soil quality in unison.

Modern soil environmental management and remediation technologies are beginning to pay attention to the long-term stability of biochar and the ecological responses of contaminants with the most toxicological fractions. Consequently, in order to meet demand and application-based results, the future application of biochar is predicted as follows:

(1) It is expected that the secondary ecological risks in the process of biochar production will still need to be focused on, i.e., the formation of HMs, VOCs, PCDD/Fs and PAHs. These pollutants are essentially derived from biomass, so it is necessary to screen and pretreat raw biomass materials to remove HMs. In addition, we must adjust the pyrolysis temperature, residence time, pyrolysis atmosphere, pressure conditions and other process parameters to minimize the generation of associated pollutants. Furthermore, it is necessary to establish and perfect the regulatory systems related to the application of biochar in practical engineering.

(2) With the passage of time, soil components will occur varying degrees of physicochemical and biochemical changes, including the properties of biochar (aged biochar) in the meantime. Although the research on soil HM immobilization by biochar has developed to laboratory pot experiments and short-term field trials, the ultimate goal is to extend it to large-scale engineering applications. Hence, carrying out long-term positioning tests in different types of HM-contaminated soil must be conducted, so as to further verify the long-term stability of biochar and its long-term effects on soil environment (such as aggregation, surface potential and density magnitude), and lastly to ensure the quality and safety of agricultural land.

(3) Biochar cannot completely remediate the heavy metals contaminated soil. Therefore, in order to improve the remediation/improvement effects of multi-heavy metals contaminated soil, the multifunctional biochar materials (e.g., biochar inoculated with microorganisms and biochar modified by chemicals and minerals) should be gradually put on the stage of engineering application.

(4) The advantages and disadvantages between the economic cost (production) and benefit value (application) of biochar need to be carefully measured. In order to enhance economic availability, easier production processes and cheaper sources of raw biomass materials need to be discovered, which could provide a platform for improving production efficiency and reducing economic burdens, i.e., to achieve the purpose of commercial practicality.

## Figures and Tables

**Figure 1 molecules-25-03167-f001:**
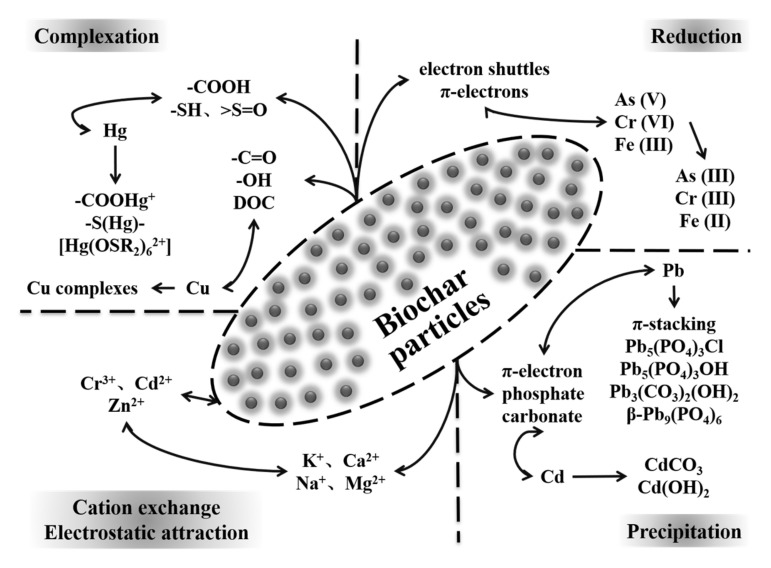
Interaction mechanism between biochar particles and HMs in soil.

**Figure 2 molecules-25-03167-f002:**
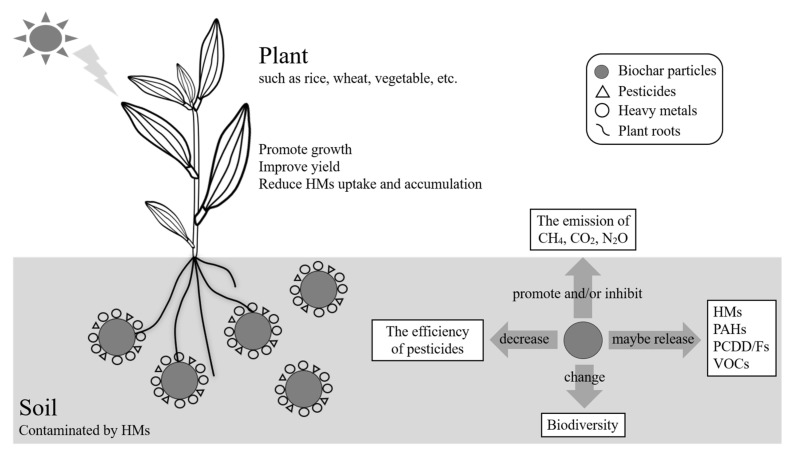
Advantages and disadvantages of biochar in the remediation of soil HM contamination.

**Table 1 molecules-25-03167-t001:** The reaction conditions and product distribution of various thermochemical conversion technologies.

Conversion Technologies	Temperature	Heating Rates	Reaction Atmosphere	Residence Time	Biochar	Bio-Oil	Syngas	Reference
Slow pyrolysis	300−650 °C	0.1−1 °C s^–1^	Oxygen-free	1−24 h	25−35%	20−30%	25−35%	[44,45]
Intermediate pyrolysis	~ 500 °C	1.0−10 °C s^–1^	Oxygen-free	10−20 s	20%	50%	30%	[46,47]
Fast pyrolysis	500−1000 °C	> 200 K min^–1^	Oxygen-free	< 2 s	12−25%	50−75%	13−25%	[48,49,50]
Gasification	750−900 °C	50−100 °C s^–1^	Air, steam, O_2_, N_2_, CO_2_ or a mixture of these gases	10−20 s	10%	5%	85%	[47,50]
Hydrothermal carbonization	180−300 °C	5−10 °C min^–1^	Confined system with a pressure of 2−6 MPa	1−16 h	50−80%	5−20%	2−5%	[19,47]
Torrefaction	250−300 °C	< 50 °C min^–1^	Inert atmosphere	10−60 min	60−80%	0%	20−40%	[44,51]

**Table 2 molecules-25-03167-t002:** Remediation efficiency of biochar on HM (heavy metal)-contaminated soil.

Metals	Types of Biomass Feedstock	Pyrolysis Temperature	Dosage	Duration ^a^	Soil Types	Total Metal Content	Immobilization Efficiency (Evaluation Method)	Reference
Cu	Chicken manure	500 °C	5%	14 d	Sedimentary alfisol	800 mg kg^−1^	73% (NH_4_NO_3_-extractable)	[36]
	Orange bagasse	500 °C	60 t ha^−1^	24 m	Fallow field soil	100 mg kg^−1^	28% (citric acid-extractable)	[90]
	Chicken manure	550 °C	5%	14 d	Mine soil	1805 mg kg^−1^	79% (NH_4_NO_3_-extractable)	[91]
	Orange bagasse	500 °C	60 t ha^−1^	6 m	Fallow field soil	100 mg kg^−1^	41% (citric acid-extractable)	[92]
	Bamboo	600 °C	15%	20 d	Sediment soil	134.6 mg kg^−1^	79.7% (HOAc-soluble)	[93]
	Oat hull	300 °C	5%	24 m	Sedimentary alfisol	338 mg kg^−1^	68% (exchangeable fraction)	[22]
	Chicken manure	550 °C	5%	14 d	Hills soil	160 mg kg^−1^	−45% (NH_4_NO_3_-extractable)	[91]
	Sewage sludge	500 °C	30 t ha^−1^	6 m	Fallow field soil	100 mg kg^−1^	−18% (citric acid-extractable)	[92]
As	Rice straw	500 °C	3%	30 d	Paddy soil	120 mg kg^−1^	As concentration increased by 234.5% in soil solution	[26]
	Rice straw	450 °C	1−3%	96 d	Paddy field soil	212 mg kg^−1^	As concentration increased in soil porewater	[94]
	Soybean stover	700 °C	10%	90 d	Agricultural soil	1945 mg kg^−1^	As mobility increased greatly in soil	[25]
	Rice straw	300 °C	10%	35 d	Paddy field soil	92.3 mg kg^−1^	As concentration increased in soil pore solution	[95]
	Oil palm fibers	700 °C	3%	20 d	Paddy field soil	0.3 mg kg^−1^	As concentration increased in soil solution	[96]
	Sewage sludge	200 °C	3%	6 d	Agricultural soil	98.7 mg kg^−1^	−81.9% (water-soluble)	[97]
	Sewage sludge	350 °C	3%	6 d	Agricultural soil	98.7 mg kg^−1^	42.2% (water-soluble)	[97]
	Corn straw	600 °C	0.5−2%	100 d	Paddy soil	73 mg kg^−1^	As(Ⅴ): 11.7−28.5% (phosphate-extractable) As(Ⅲ): 54.0−81.6% (phosphate-extractable)	[23]
Cd	Wheat straw	350−550 °C	40 t ha^−1^	3 y	Paddy soil	5 mg kg^−1^	59% (CaCl_2_-extractable) 24% (DTPA-extractable)	[98]
	Bamboo	600 °C	15%	20 d	Sediment soil	3.8 mg kg^−1^	31.2% (HOAc-soluble)	[93]
	Soybean straw	350 °C	3%	6 d	Agricultural soil	1.36 mg kg^−1^	65.7% (water-soluble)	[97]
	Maize straw	550 °C	30 t ha^−1^	~ 6 m	Paddy soil	2.04 mg kg^−1^	50.4% (DTPA-extractable)	[29]
	Rice straw	450 °C	1−2%	96 d	Paddy field soil	10.8 mg kg^−1^	Cd concentration decreased in soil porewater	[94]
	Sugarcane bagasse	500 °C	1.5%	4 m	Agricultural soil	50 mg kg^−1^	40.4% (DTPA-extractable)	[21]
	Corn stalk	550 °C	2%	30 d	Arable land soil	2.0 mg kg^−1^	91% (CaCl_2_-extractable)	[99]
	Hickory nut shell	500 °C	30 t ha^−1^	~ 6 m	Paddy soil	2.04 mg kg^−1^	53.6% (DTPA-extractable)	[29]
Pb	Soybean stover	700 °C	10%	90 d	Agricultural soil	1945 mg kg^−1^	95% (NH_4_OAc-extractable) 95% (TCLP-extractable)	[25]
	Vegetable waste	500 °C	5%	45 d	Agricultural soil	1445 mg kg^−1^	87% (NH_4_OAc-extractable)	[100]
	Bamboo sawdust	600 °C	37.5%	30 d	Sediment soil	589.7 mg kg^−1^	100% (TCLP-extractable)	[101]
	Red pepper stalk	650 °C	2.5%	45 d	Agricultural soil	1445 mg kg^−1^	65% (NH_4_OAc-extractable)	[100]
	Wheat straw	350−550 °C	40 t ha^−1^	3 y	Paddy soil	100 mg kg^−1^	59% (CaCl_2_-extractable) 27% (DTPA-extractable)	[98]
	Bamboo	600 °C	15%	20 d	Sediment soil	44.3 mg kg^−1^	73.2% (HOAc-soluble)	[93]
Hg	Rice husk	550 °C	1−5%	10 d	Field soil	1000 mg kg^−1^	> 94% (TCLP-extractable)	[102]
	Rice hull	480−660 °C	24 t ha^−1^	118 d	Farmland soil	129 mg kg^−1^	Hg concentration decreased by 44% in soil porewater	[28]
	Wheat straw	350−450 °C	72 t ha^−1^	118 d	Farmland soil	129 mg kg^−1^	Hg concentration decreased by 26% in soil porewater	[28]
	Sewage sludge	600 °C	5%	17 w	Paddy field soil	2.1 mg kg^−1^	MeHg concentration increased by 67% in soil	[41]
	Sewage sludge	600 °C	5%	17 w	Paddy field soil	65.3 mg kg^−1^	MeHg concentration increased by 29% in soil	[41]
Cr	Waste wood	900 °C	1−5%	11 w	Tannery waste soil	12285 mg kg^−1^	28−68% (CaCl_2_-extractable)	[103]
	Sugarcane bagasse	500 °C	1.5%	4 m	Agricultural soil	50 mg kg^−1^	49.6% (DTPA-extractable)	[21]
	Rice straw	500 °C	40 t ha^−1^	~ 4 m	Paddy field soil	432.8 mg kg^−1^	Cr(total): 48.1% (HNO_3_/H_2_SO_4_-extractable) Cr(Ⅵ): 22.3% (HNO_3_/H_2_SO_4_-extractable)	[104]
	Wheat straw	600 °C	0.25%	180 d	Cr-spiked soil	308 mg kg^−1^	Cr(Ⅵ): 47.1% (TCLP-extractable) Cr(Ⅵ): 65.5% (CaCl_2_-extractable)	[105]

^a^ Units explanation: d for days, w for weeks, m for months and y for years.

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
