# Peer review of "Application Research of Biochar for the Remediation of Soil Heavy Metals Contamination: A Review"

_molecules, 2020, doi:10.3390/molecules25143167_

Round 1

Reviewer 1 Report

This paper summarizes the literature on the use of biochar produced by pyrolysis to reduce the heavy metal (HM) pollution of soil and, ultimately, food, as the paper notes.

The summary of the literature on the benefits in the reduction of HM pollution is good, and the paper is somewhat cautious especially about the production of greenhouse gases, where some literature just asserts that greenhouse gases are not a problem with biochar.

Two further points need a little more discussion.  One is that earthworms, fungi, and fertilizer are essential to food production, and biochar can interfere with all of them.  The paper says that.  Is there any literature on the effect of biochar on the total production of uncontaminated food?  If not, it is important to state that.

On line 430, the "problematic production process" of biochar is mentioned, but not described in any detail at all.  The high temperatures needed to produce biochar require fuel.  This is a major problem with desalination:  fresh water from sea water appears to be a great idea, but the energy needed is very large, with significant greenhouse gas emissions.  So is biochar subject to a similar problem?

Author Response

Please see the attachment. The yellow part of the manuscript indicates that it has been modified or may have been modified.

Reviewer 2 Report

This review paper provides an overview of the application of biochar as an amendment for the remediation of heavy metals contaminated soil. This topic is very hot in recent years and within the scope of the journal. The review content is comprehensive, which is helpful for other researchers working in the same areas. Overall, the writing is commendable and the presentation of the content is very clear. However, more details should be provided in some parts of the paper. The reviewer has some comments and suggestions to help the authors to further improve the quality of the review manuscript. If the authors are willing to revise, the manuscript can be considered for publication in Molecules. The general and details comments are summarised as follows.

  1. When discussing the interactions between biochar and soil heavy metals, the mobility of heavy metals in unsaturated contaminated soil is an important factor that affects the biochar-HM interactions. Soil hydraulic properties affect the mobility of heavy metals. Hence, the influence of biochar on soil hydraulic properties (water retention and hydraulic conductivity) should be also reviewed. Even though authors conclude that biochar can improve soil water holding capacity in the Abstract, but there is no content about this in the main text. Regarding the biochar effects on soil hydraulic properties, the authors can refer to the following articles “Effects of biochar on water retention and matric suction of vegetated soil”, “Two-year evaluation of hydraulic properties of biochar-amended vegetated soil for application in landfill cover system”, “Influence of biochar addition on gas permeability in unsaturated soil” and others.
  2. For the future perspectives, the authors should extend this section by including the possible methodologies to achieve those proposed future topics. The current last paragraph is too brief. In addition, the measures to mitigate the negative effects of biochar on soil environment and heavy metal remediation should also be discussed in detail.
  3. In each subsection of Section 4“Remediation of soil HMs contamination by biochar”, more details should be provided regarding the time efficiency of each heavy metal remediation by using different biochar types.
  4. In section “3.4. pH values”, only alkaline biochar is reviewed. Actually, some biochars can be acidic. Please also review and discuss the acidic biochar and its influence on soil pH as well as soil heavy metal remediation.
  5. What are the typical ranges of biochar particle size produced at different pyrolysis temperatures? What are the effects of biochar particle size on soil properties and biochar-heavy metal interactions?
  6. L155: There are two “changed”. One should be deleted.

Author Response

(The authors gave the same response as above.)

Round 2

Reviewer 2 Report

The authors have addressed my concerns satisfactorily. The paper can be accepted for publication now.